# Crowd Evacuation in Hajj Stoning Area: Planning through Modeling and Simulation

**Heba Kurdi** [1,2,*], **Amal Alzuhair** [1], **Dana Alotaibi** [1], **Hesah Alsweed** [1], **Noor Almoqayyad** [1], **Razan Albaqami** [1], **Alhanoof Althnian** [3], **Najla Alnabhan** [1] and **A. B. M. Alim Al Islam** [4]

1   Computer Science Department, King Saud University, Riyadh 11451, Saudi Arabia; 436200038@student.ksu.edu.sa (A.A.); 437200118@student.ksu.edu.sa (D.A.); 437201650@student.ksu.edu.sa (H.A.); 442204436@student.ksu.edu.sa (N.A.); razanalbaqami@gmail.com (R.A.); nalnabhan@ksu.edu.sa (N.A.)
2   Department of Mechanical Engineering, Massachusetts Institute of Technology (MIT), Cambridge, MA 02139, USA
3   Information Technology Department, King Saud University, Riyadh 11451, Saudi Arabia; aalthnian@ksu.edu.sa
4   Department of Computer Science and Engineering, Bangladesh University of Engineering and Technology, Dhaka 1000, Bangladesh; alim_razi@cse.buet.ac.bd
*   Correspondence: hkurdi@ksu.edu.sa

**Abstract:** Pilgrimage is one of the largest mass gatherings, where millions of Muslims gather annually from all over the world to perform Hajj. The stoning ritual during Hajj has been historically vulnerable to serious disasters that often cause severe impacts ranging from injuries to death tolls. In efforts to minimize the number and extent of the disasters, the stoning area has been expanded recently. However, no research has been carried out to study the evacuation effectiveness of the current exit placements in the area, which lies at the heart of effective minimization of the number and extent of the disasters. Therefore, this paper presents an in-depth study on emergency evacuation planning for the extended stoning area. It presents a simulation model of the expanded stoning area with the current exit placement. In addition, we suggested and examined four different exit placements considering evacuation scenarios in case of no hazard as well as two realistic hazard scenarios covering fire and bomb hazards. The simulation studied three stoning phases, beginning of stoning, during the peak hour of stoning, and ending of stoning at three scales of population sizes. The performance was measured in the light of evacuation time, percentage of evacuees, and percentage of crowd at each exit. The experimental results revealed that the current exits are not optimally positioned, and evacuation can be significantly improved through introducing a few more exits, or even through changing positions of the current ones.

**Keywords:** facilities planning and design; decision support; crowd planning; evacuation; simulation; pilgrimage

## 1. Introduction

There exist many notable examples of human crowds all over the world, where crowd disasters are not uncommon. As a direct result of the increased number of disasters in human crowds in recent times, researchers' interest in evacuation planning has increased. Planning is crucial since areas that are not well prepared with an emergency evacuation plan can be easily threatened by serious dangers with lethal consequences such as fires and communicable hazards.

The Muslim pilgrimage, Hajj, is the world's largest annual human gathering. It takes place in Makkah, Saudi Arabia. Every year, over two million Muslims from across the world gather in one place at the same time, and, except for the last two years, the number is constantly growing [1]. The large congregations of this many pedestrians in a relatively

small area have made Hajj historically hazardous. While crowd management has improved significantly in recent years, the Hajj remains susceptible to disasters from crowding.

The stoning area in Makkah experiences an important ritual, where all pilgrims need to perform the stoning ritual within a limited time. This area has already witnessed several deadly incidents over the past decades [1,2]. Therefore, it demands a carefully crafted emergency crowd evacuation plan as well as other developments, such as an expansion of the area and changes in the placements of exits as needed.

In recent times, the stoning area has witnessed several expansions projects by the Hajj authority. The overall pursuit of the recent expansions of the stoning area as preparation for the turnout of millions of pilgrims has yet to be free from fears of repeating disaster incidents. This happens as, despite the expansions, corresponding emergency crowd evacuation planning is yet to be investigated in depth. A few of the previous research studies have focused on evacuation plans considering the previous (not-expanded) stoning area, which is no more applicable.

The latest efforts of the Hajj authority included the area's expansions to enable a capacity of containing more pilgrims and adopting different crowd management plans, which have drastically changed the current contexts. Hence, it demands a new and in-depth study on emergency crowd evacuation planning over the stoning area. To the best of our knowledge, such a study is yet to be performed in the literature. This work seeks to fill this gap.

Accordingly, this paper presents an in-depth study on the emergency crowd evacuation planning over the stoning area in Makkah considering its recent expansions. To do so, we simulated the crowd dynamics of pedestrians (pilgrims) in the area potentially under different types of disasters. We focused on the second floor of the stoning area and developed a simulation model to study the crowd evacuation planning during an emergency given the current exit layout, as well as newly-proposed exits. Further, we evaluated the effectiveness of the current and proposed exit plans taking into account the three performance metrics: evacuation time, percentage of evacuees, and percentage of crowds at each exit, and different hazard scenarios, including fires and bombs.

The main contributions of the paper can be summarized as follows:

- Building a simulation model of the expanded stoning area of Jamarat to analyze the exit capacity of the current settings while having no hazards.
- Considering different types of potential hazard scenarios (fires and bombing) to further analyze the evacuation capability of the extended area under hazards.
- Proposing new exit plans with different exit placements in the expanded area.
- Testing the different exit placements to evaluate the effectiveness of emergency evacuation plans in terms of evacuation time, percentage of evacuees, and percentage of crowd at each exit.

The rest of the paper is organized as follows. Section 2 reviews the related work. In Section 3, the model design is described. Section 4 presents the evaluation methodology and discusses the simulation results. Finally, Section 5 concludes the paper.

## 2. Literature Review

Pilgrimage is the world's largest annual human gathering, where millions of people from all over the world gather in a limited space during a specific period to perform religious rituals. As recent decades have witnessed many deadly incidents, numerous efforts have been made to manage and model pilgrimage crowd. In the text below, we highlight the main efforts made in this field, and we summarize the most related works in Table 1.

**Table 1.** Summary of related work.

| Reference | Goal | Model | Performance Measures |
|---|---|---|---|
| [1] | Crowd management | A non-linear congestion network model | The sum of the non-linear congestion measure overall routes and sites. |
| [2] | Crowd modeling: Hajj education | Agent-based | Survey to test if the model can help people learn about how to perform pilgrimage and evaluate the accuracy of crowd simulation. |
| [3] | Crowd simulation | 3D model | Empirical and user testing to evaluate the usability and functionality of the designed system. |
| [4] | Crowd modeling | Cellular automata and discrete events | Comparing speed-density graphs obtained from empirical observations. |
| [5] | Crowd modeling and evacuation | Agent-based | Number of pilgrims in queue for different shapes of stoning pillars, different ranges of stoning, different queue thresholds, different ticks, and time to do stoning. |
| [6] | Crowd management | Stochastic Data-traffic networks | Evacuation time, average travel time from Arafat to Muzdalifah, and congestion level. |
| [7] | Crowd modeling and evacuation | Agent-based | Evacuation time was measured through the mean plots for three types of evacuation strategies and a total of five distinct scenarios. |
| [8] | Evacuation | Cellular automata | Chosen path, evacuation time, and percentage of crowds at exits. |
| [9] | Crowd management | N/A | Accuracy of head-count Algorithms to compute the likelihood of stampedes. |
| [10] | Crowd management | N/A | Survey on different possible technologies for Crowd management and comparison over them |
| [11] | Crowd monitoring | Agent-based and regression model | Speed and travel time in Ziara |
| [12] | COVID spreading | Agent-based | Spread of infection during Tawaf and Ramy al-Jamarat |
| [13] | Crowd modeling | Discrete event | Evacuation duration and number of evacuated groups in Tawaf and Sayee |

In [1], a non-linear temporal network model was introduced to represent the traffic flow in holy sites, i.e., Makkah and Medina, during Hajj. The model focused on religious, spatial, and time constraints to ease the congestion. The model provided several suggestions to assist in alleviating the overcrowding problem and provide insights for planning for expansions in holy sites' routes. Apart from evacuation strategies, other studies tried to implement the concept of the intelligent agent such as in [2]. Agents in their simulation model were able to adapt to the environment and can make decisions with the available knowledge. The model can be used for training pilgrims prior to performing the ritual. To

evaluate the model, the authors conducted a questionnaire which resulted in confirming that the system is beneficial to learn pilgrimage rituals. Similarly, the work in [3] proposed a training method with an interactive and flexible user interface. The authors presented a 3D simulation for the Kaaba circumambulation using COLLADA model supported with the 3D game engine Horde. The authors in [4], focused on crowd modeling, where they presented a model for pedestrian movements and used discrete event techniques to simulate the crowd's behavior. In addition, they used circular cellular automata to model Kaaba circumambulation.

A model for the devil stoning ritual was developed in [5], where a conceptual design was created to analyze safety parameters such as evacuation strategies and peak load. A general-purpose agent was developed to study the effect of various behaviors or parameters on the agents performing pilgrimage. Agents' behaviors such as aggressiveness or walking pace were studied. In [6], the authors developed a model to enable real-time traffic management for the pilgrimage authorities. They used modeling techniques, such as adaptive control, to simulate Makkah's road networks and achieve effective traffic awareness between the holy sites.

Aiming to produce a model to analyze and optimize situations of huge mass gatherings, the work in [7] developed a model based on simulating crowded agents using Anylogic simulation tool. To evacuate agents in the model, the authors used three different evacuation approaches, namely Random Crowd Evacuation, Shortest Regional Distance (SRD), and Genetic Algorithm, and evaluated them on a population of 10,000 agents. Their results showed that SRD outperformed the Random Crowd evacuation. In their specified measures, the Genetic algorithm gave the best result although it starts slower. In [8], researchers tested the effect of placements of exits by designing an evacuation system. Four different exit placements were tested to determine the most efficient one in case of evacuation. To test the scenarios, the authors used simulated annealing (SA) and depth-first search (DFS) and evaluated the system considering the percentage of crowds and evacuation time at each exit. Their results showed that placing exits on the opposite side of the main exit gives the best performance. The authors in [9] presented a framework for crowd control and discussed real-life cases of pilgrimage. The framework has two subsystems: Disaster Control and Management Systems (DCMS) and Healthcare Management System (HMS). In their work, the authors developed an integrated mobile application and proposed several techniques to predict stampedes. Further, they used cloud computing, Wireless Sensor Networks (WSNs), Global Positioning System (GPS), and Internet Protocol (IP) cameras to enhance the efficiency of the application.

Furthermore, there exist some recent related studies. For example, the study in [10] surveys digital solutions for Hajj crowd management while [11] presents a simulation-based analysis for flow control and crowd monitoring in Ziara. The study in [12] presents agent-based modeling on the possible spreading of COVID-19 during Hajj Rituals. Another study models crowd through discrete event simulation covering Tawaf and Sayee Rituals during Hajj [13]. However, similar to all the earlier studies, none of these studies has focused on crowd evacuation in the extended stoning area through efficient exit planning that we do in this study.

In summary, the existing studies are yet to develop a model of the expanded stoning area of Jamarat that can facilitate analyzing the exit capacity of the current settings while having. Furthermore, the existing studies are yet to analyze the evacuation capability of the extended area under different types of potential hazard scenarios such as fires and bombing. Additionally, proposing new exit plans with different exit placements in the expanded area remains yet another unexplored area in the literature. Further, in the process of the exploration of the exit placements, it is extremely important to test different exit placements to analyze the effectiveness of emergency evacuation plans in terms of different performance metrics such as evacuation time, percentage of evacuees, and percentage of crowd at each exit. However, such testing is yet to be attempted in the literature. Therefore,

in this study, we attempt to fill the gaps in the literature through exploring all these unexplored areas.

## 3. Model Design

The aim of this research covers helping in improving crowd evacuation in Hajj Stoning areas and examining factors that might affect the safe evacuation process. We considered the second floor of the stoning area as a case study since it is part of the most recent extension project and has not been modeled before [14]. The simulation models were developed using the Netlogo simulation tool [15] which is an agent-based simulation environment that allows true heterogeneity in simulating the motion and trajectory of each individual. In the following sub-section, we explain each model in detail.

### 3.1. Environment Model

The simulation environment consists of square cells, where each cell corresponds to a square meter in the real-world [16] and is occupied by only one agent. As mentioned earlier, this model focuses on the second level of the stoning area, which is 950 m long and 80 m wide with a single entrance and two exits [17]. An illustration of the stoning area is shown in Figure 1a.

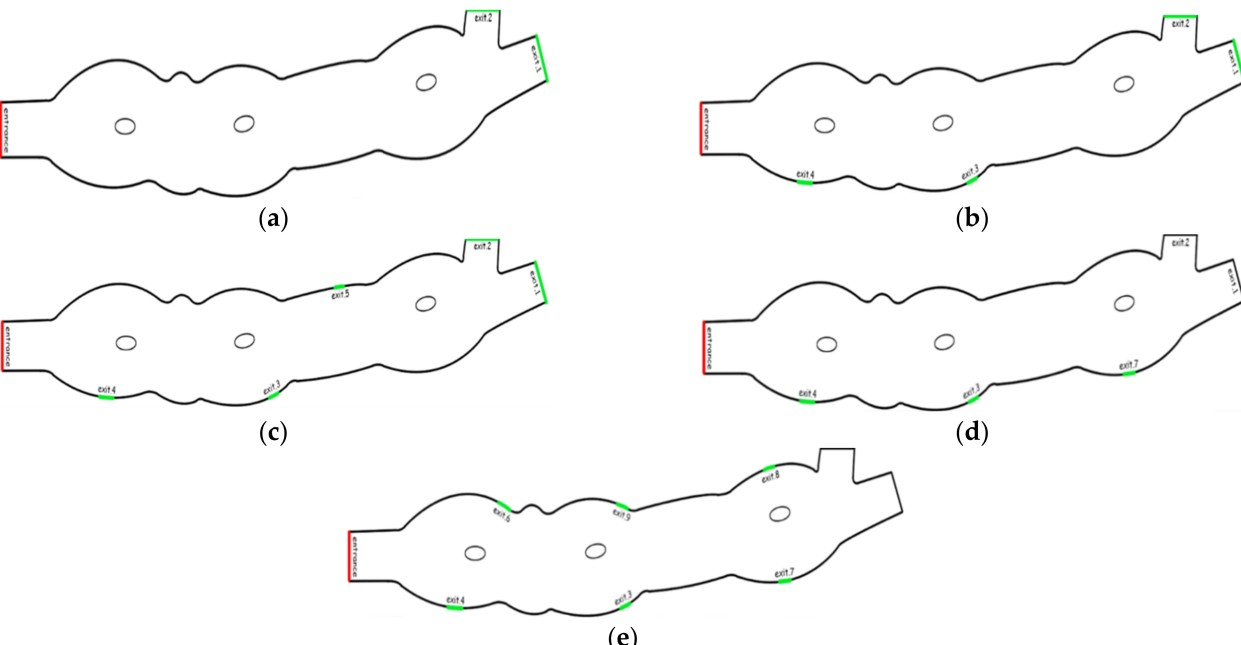

**Figure 1.** Illustrations of the current and proposed exit placements. (**a**) Current exits. (**b**) Current and two adjacent exits. (**c**) Current and three opposite exits. (**d**) Three adjacent exits. (**e**) Six opposite exits.

In our work, and as shown in the figure, the model is defined with white walls, green exits, and red entrance. Since exits placement has a dominant effect on evacuation plans [8], we propose and test several exit placements and compare them with the current exit placement to investigate if evacuation can be improved in the area. The proposed exit placements are depicted in Figure 1b–e, which vary based on the number and position of the exits, including (1) current and two adjacent exits (Figure 1b), (2) current, and three opposite exits (Figure 1c), (3) three adjacent exits (Figure 1d), and (4) six opposite exits (Figure 1e).

Note that the proposed exit placements come from the experiences of the researchers. Most of the authors of this paper [18] are from the Saudi background and have a comprehensive idea of the overall area of the stoning under consideration. All the proposed exit placements are engendered from their ideas and experiences through mutual discussions and brainstorming.

### 3.2. Agent Model

Pilgrims are the main agents in our simulation. Since pilgrims differ in many aspects, we considered a general case of agents' behaviors focusing only on the speed of agents based on the typical demographics of pilgrims and the decision of choosing an exit. Several parameters of agent population were considered including agents' population size, agent distribution, and exit trajectory. Three increasing values of the agent population size were considered: 3000, 7000, and 10,000 agents. Pilgrim agents were distributed in the area according to three phases of stoning, namely (1) beginning of the stoning ritual, (2) peak hours of the ritual, and (3) the end of the ritual, as shown in Figure 2. At the beginning of the stoning ritual, agents arrive at the area to perform the ritual (Figure 2a). During peak hours, agents continue to arrive at the area, while some agents are performing the ritual (Figure 2b). Lastly, at the end of the ritual, pilgrims are no longer entering the bridge, but rather only exiting after finishing the ritual (Figure 2c).

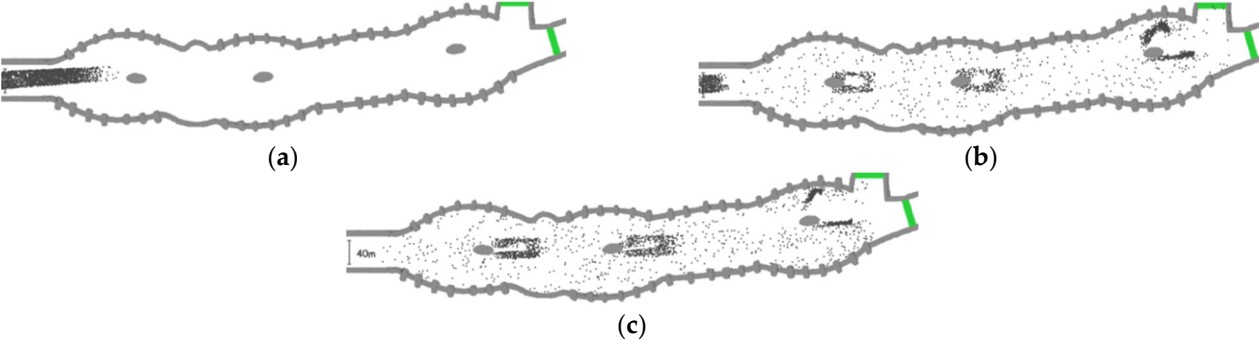

**(a)**          **(b)**

          **(c)**

**Figure 2.** Agents distribution techniques. (**a**) At the beginning of the ritual. (**b**) During peak hours of the ritual. (**c**) At the end of the ritual.

Speed values were assigned using the normal distribution to capture the walking speeds of pilgrims, which vary from slow walking (0.457 m/s) to slow running (1.676 m/s) [19]. Pilgrims' age falls in the range 18–94 years old, with an average age of 61 years old years. Therefore, the standard deviation of the normal distribution was set to 0.198 m/s, and hence most pilgrims walk at a slow speed [20]. Further, pilgrims' movements were defined by other factors which are the agent's vision and direction. Figure 3 shows an agent's vision range, which is limited to five cells ahead of the agent. These cells define the locations where the agent can move to. Lastly, exit trajectory was used to guide the agents' movements towards exits. During the simulation, each pilgrim agent leaves the area using the closest exit. At each step, the agent considers the five visible cells and moves toward the cell that is closest to the selected exit.

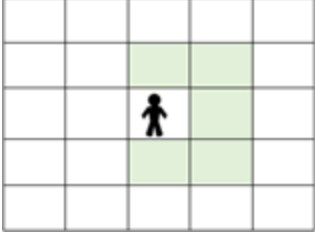

**Figure 3.** Agent's vision range.

### 3.3. Hazard Models

Two hazard models were considered in this work, fire condition, and bomb attack. Similar to [21], the fire starts at a fixed location and expands in all directions every 5–10 s until the end of the simulation (i.e., all alive agents evacuated). The bomb attack starts at the most crowded area depending on when it occurs (beginning of stoning ritual, peak

hours of the ritual, or at the end of the ritual). Similar to [22], a bomb explosion lasts for 20 s, and it impacts pedestrians in the proximity of 50 meters. Simulation parameters of the hazard models are presented in Table 2.

**Table 2.** Fire and Bombing models' parameters.

| Hazard | Starting Location | Spread Rate | Spread Direction | Impact | Area Coverage | Duration |
|--------|-------------------|-------------|------------------|--------|---------------|----------|
| Fire | Fixed | 5–10 s | All | Kills pilgrims on contact | Entire area | Until end of simulation |
| Bomb | Mostcrowded | NA | All | Kills pilgrims in 50 $m^2$ proximity | 50 $m^2$ | 20 s |

## 4. Model Evaluation and Simulation Results

This section presents the evaluation methodology and results. Section 4.1 describes the simulation settings and performance metrics while Section 4.2 illustrates the results for each metric.

### 4.1. Simulation Settings and Performance Metrics

The model focuses on the effect of exits placement under different factors such as hazards to obtain the best evacuation plan for the stoning area. We tested five different exit placements in the stoning area including the current placement. Further, we varied the hazard types, crowd distribution, and pilgrims' population size. Table 3 presents the different values assigned for each parameter.

**Table 3.** Variables and their assigned values.

| Variable | Assigned Values |
|----------|-----------------|
| Exits Placement | Current exits |
| | Current and two adjacent exits |
| | Current and three opposite exits |
| | Three adjacent exits |
| | Six opposite exits |
| Hazard Type | None |
| | Fire |
| | Bomb |
| Crowd Distribution | At the beginning of the ritual |
| | During peak hours of the ritual |
| | At the end of the ritual |
| Population size | 3000 |
| | 7000 |
| | 10,000 |

Three performance metrics are measured in this work, namely evacuation time (ET), percentage of evacuees (PE), and percentage of crowd at an exit (PCE). Below, is the description of each metric:

- Evacuation Time: This metric indicates the average evacuation time of all pilgrims. For each pilgrim agent, evacuation time represents the duration of time it takes to leave the stoning area using the closest exit.

- Percentage of Evacuees: This metric represents the percentage of pilgrims who successfully left the stoning area during the evacuation time, according to Equation (1).

$$\frac{v \times 100}{N} \tag{1}$$

where:

$v$ is the total number of evacuees, and
$N$ is the total number of pilgrim agents.

- Percentage of Crowd at an Exit: This metric is used to investigate the effectiveness of exits placement. It is computed for each exit as the percentage of pilgrim agents that used the exit to leave the area, as shown in Equation (2):

$$\frac{P_i \times 100}{N} \tag{2}$$

where:

$P_i$ is the number of pilgrim agents that left using exit $i$, and
$N$ is the total number of pilgrim agents.

Each scenario was repeated three times and the mean outcomes were computed to improve the accuracy of the reported results and reduce possible variability resulting from randomization.

### 4.2. Results on Evacuation Time

The performance in respect to evacuation time is shown in Figures 4–6 with no hazard, fire hazard, and bomb hazard, respectively. In each figure, a sub-figure presents the evacuation time for each exit placement considering a specific phase of stoning and a certain pilgrims' population size.

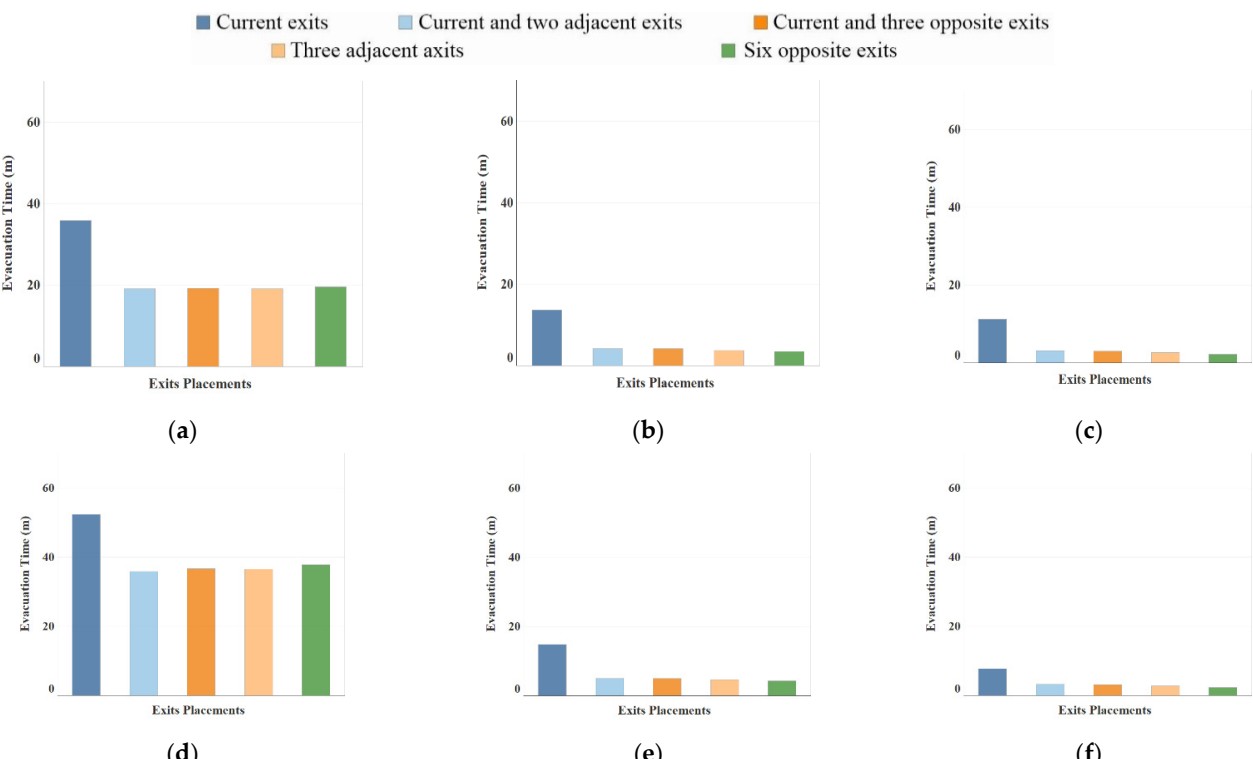

**Figure 4.** *Cont.*

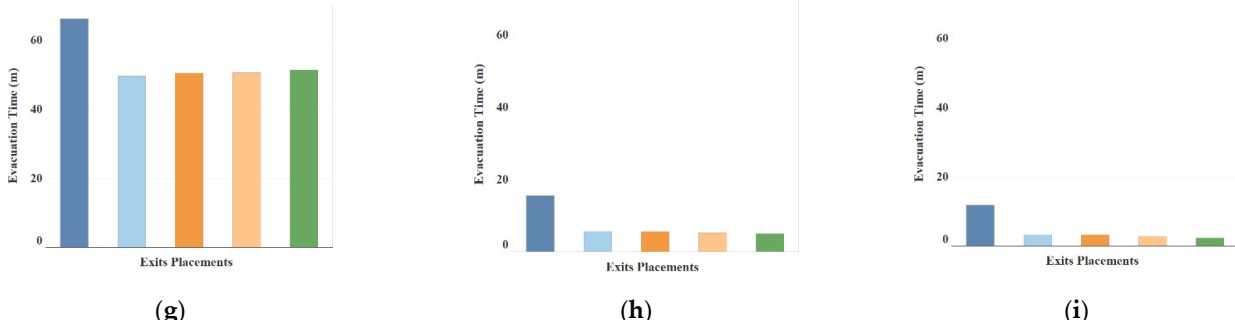

**Figure 4.** Average evacuation time for all scenarios in case of no hazard with 3000 pilgrims (**a**–**c**), 7000 (**d**–**f**), and 10,000 pilgrims (**g**–**i**). (**a**) Beginning of stoning with 3000 pilgrims. (**b**) During peak hour of stoning with 3000 pilgrims. (**c**) End of stoning with 3000 pilgrims. (**d**) Beginning of stoning with 7000 pilgrims. (**e**) During peak hour of stoning with 7000 pilgrims. (**f**) End of stoning with 7000 pilgrims. (**g**) Beginning of stoning with 10,000 pilgrims. (**h**) During peak hour of stoning with 10,000 pilgrims. (**i**) End of stoning with 10,000 pilgrims.

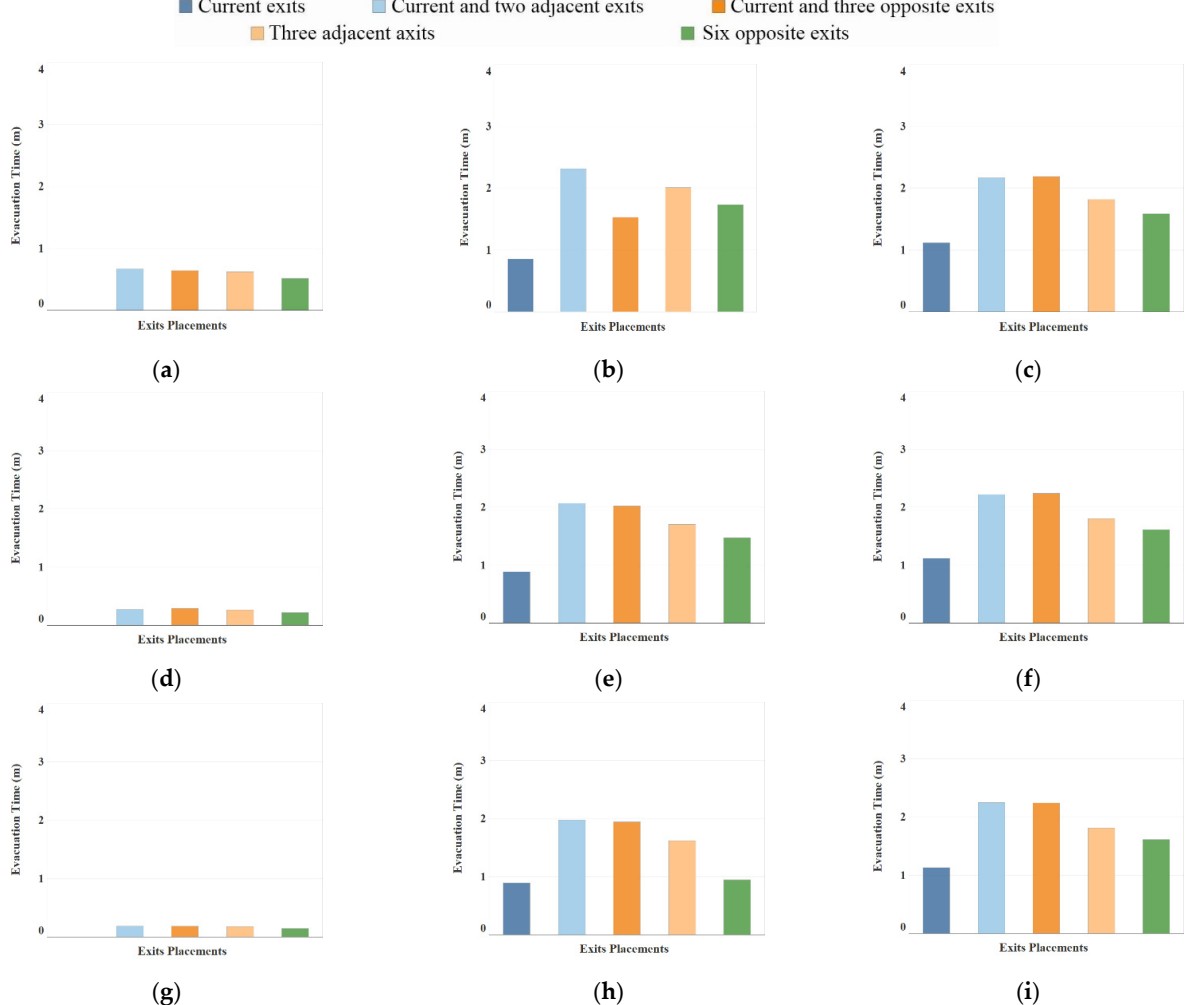

**Figure 5.** Average evacuation time for all scenarios in case of fire with 3000 pilgrims (**a**–**c**), 7000 pilgrims (**d**–**f**), and 10,000 pilgrims (**g**–**i**). (**a**) Beginning of stoning with 3000 pilgrims. (**b**) During peak hour of stoning with 3000 pilgrims. (**c**) End of stoning with 3000 pilgrims. (**d**) Beginning of stoning with 7000 pilgrims. (**e**) During peak hour of stoning with 7000 pilgrims. (**f**) End of stoning with 7000 pilgrims. (**g**) Beginning of stoning with 10,000 pilgrims. (**h**) During peak hour of stoning with 10,000 pilgrims. (**i**) End of stoning with 10,000 pilgrims.

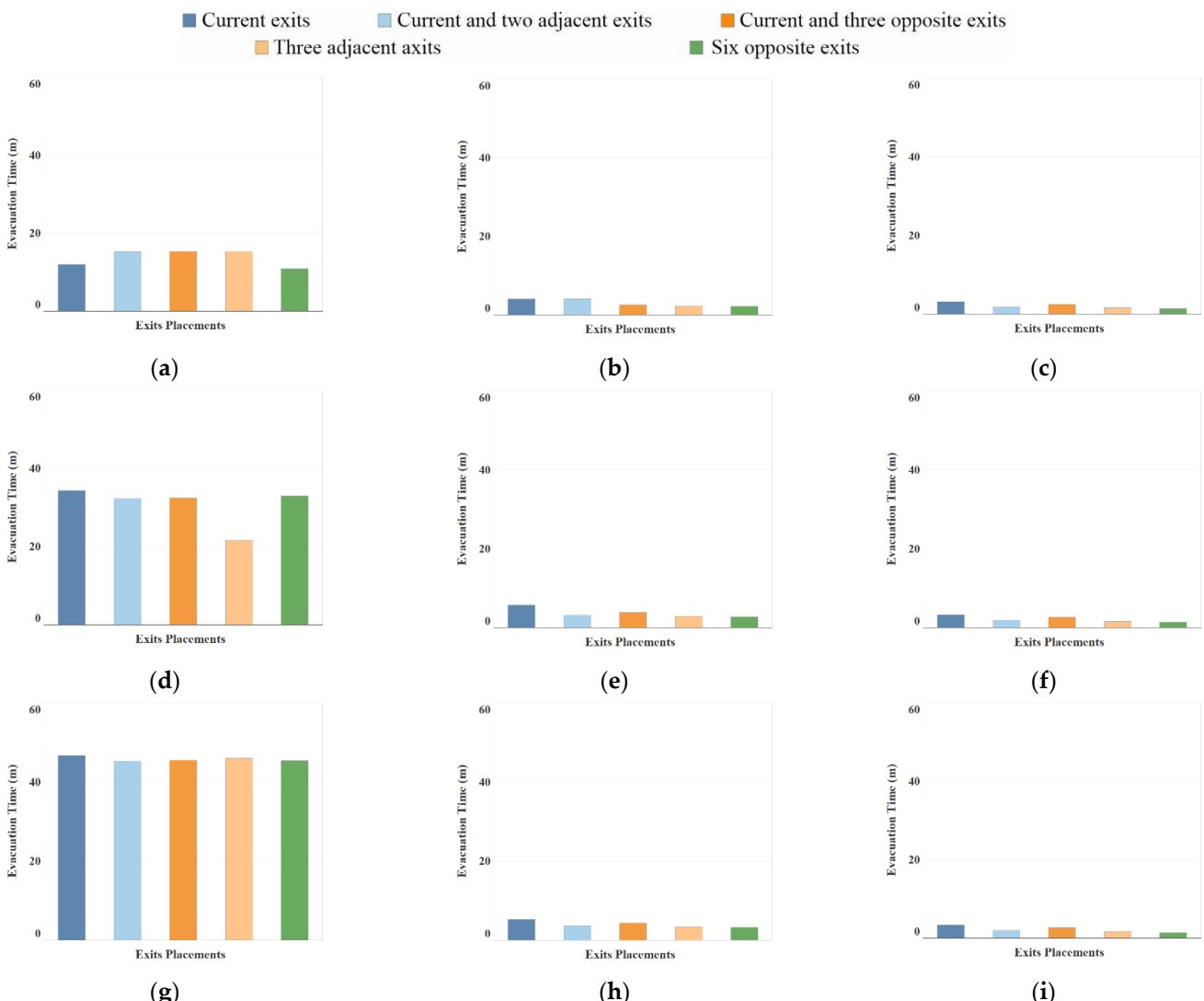

**Figure 6.** Average evacuation time for all scenarios in case of bomb with 3000 pilgrims (**a**–**c**), 7000 pilgrims (**d**–**f**), and 10,000 pilgrims (**g**–**i**). (**a**) Beginning of stoning with 3000 pilgrims. (**b**) During peak hour of stoning with 3000 pilgrims. (**c**) End of stoning with 3000 pilgrims. (**d**) Beginning of stoning with 7000 pilgrims. (**e**) During peak hour of stoning with 7000 pilgrims. (**f**) End of stoning with 7000 pilgrims. (**g**) Beginning of stoning with 10,000 pilgrims. (**h**) During peak hour of stoning with 10,000 pilgrims. (**i**) End of stoning with 10,000 pilgrims.

In case of no hazard, as can be seen in Figure 4, pilgrims take more time to evacuate at the beginning of the stoning ritual, compared to peak hour and end of ritual phases. This applies to all exit placements and pilgrim population sizes. The reason is probably that, at the beginning of the ritual, pilgrims are still arriving at the stoning area and hence are mostly located near the entrance. Therefore, pilgrims take more time to evacuate since they travel more distance, compared to peak hours and end of ritual phases.

On the contrary, if a fire occurs at the beginning of stoning (Figure 5a,d,g), pilgrims can evacuate faster than the corresponding scenarios at peak hour (Figure 5b,e,h) and end of stoning (Figure 5c,f,i). As mentioned earlier, in our simulation, a fire occurs at cell (9643, 109), which is close to the entrance, spreads every 5–10 s, and kills pilgrims on contact. Therefore, since at the beginning of stoning most pilgrims are entering the area and are close to the entrance, it is perhaps the case that most pilgrims are killed by the fire and only a few survive and evacuate in a short time.

In the bomb scenario, as shown in Figure 6, and similar to the no-hazard scenario, it can be observed that pilgrims take more time to evacuate at the beginning of stoning, compared to peak hour and end of stoning phases. Recall that in our simulation a bomb explodes at the most crowded area for only 20 seconds and kills pilgrims at the proximity of 50 m$^2$. Consequently, the number of pilgrims who are killed by a bomb explosion is probably similar in all hazard scenarios since it targets the most crowded area. As a result, the bomb scenario has similar evacuation time trends to the no-hazard scenario across stoning phases.

When comparing the placement of the current exits with the proposed placements in the three hazard scenarios in Figures 4–6, it can be observed that the current exit placement has a longer evacuation time in the no hazard scenario, shorter evacuation time in the fire hazard, and almost similar evacuation time in the bomb hazard. Further, all proposed exit placements have almost similar evacuation times in the no-hazard and bomb scenarios (Figures 4 and 6). In all scenarios, as expected, the larger the population size, the longer is the evacuation.

### 4.3. Results on Percentage of Evacuees

The performance in respect to the percentage of evacuees is shown in Figures 7 and 8 with fire and bomb hazards, respectively. The no-hazard scenario was not considered since it is expected that all pilgrims evacuate successfully. In each figure, a sub-figure presents the percentage of evacuees for each exit placement considering a specific stoning phase and population size.

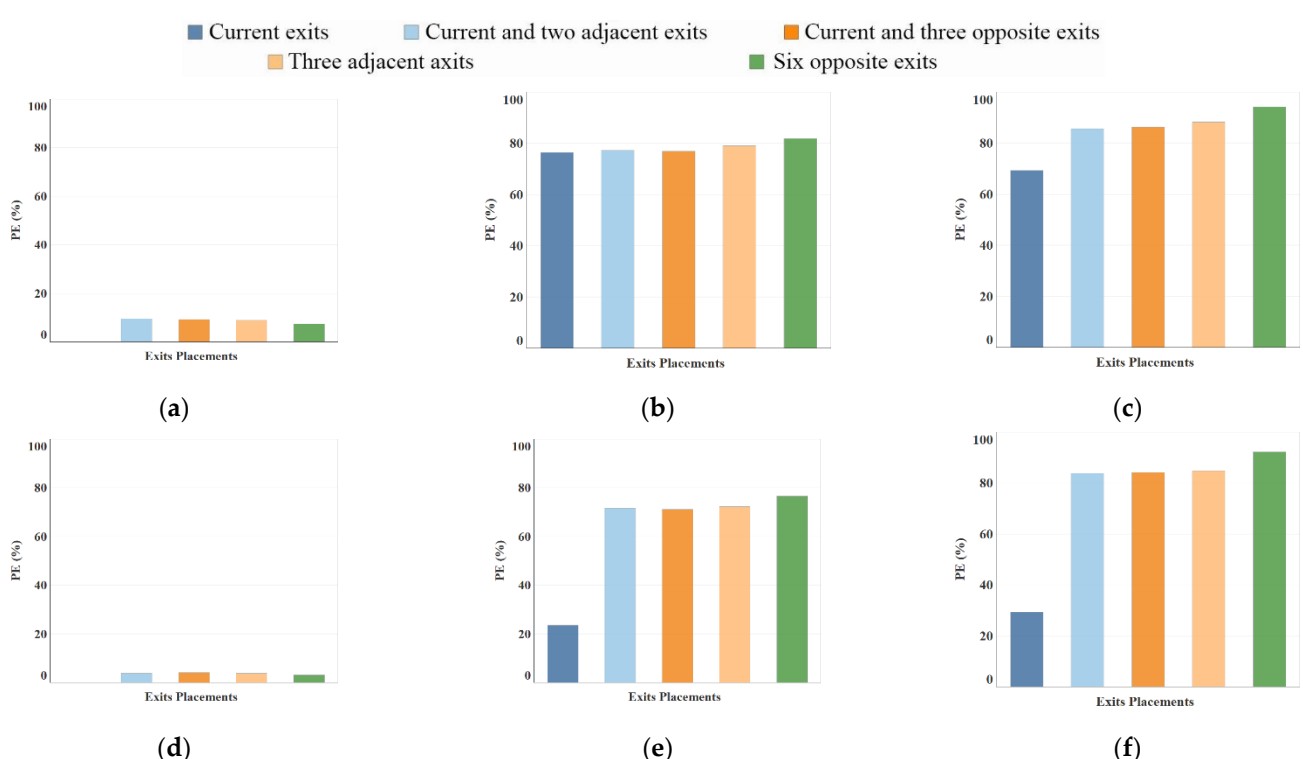

**Figure 7.** *Cont.*

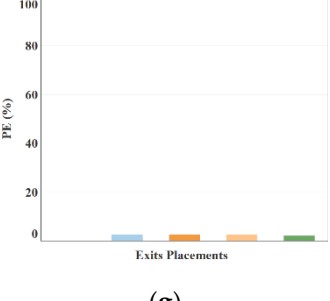

(**g**)

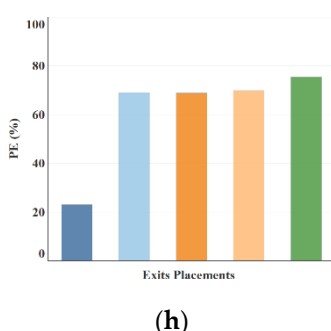

(**h**)

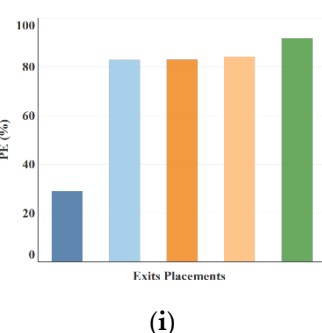

(**i**)

**Figure 7.** Percentage of evacuees (PE) for all scenarios in case of fire with 3000 pilgrims (**a**–**c**), 7000 pilgrims (**d**–**f**), and 10,000 pilgrims (**g**–**i**). (**a**) Beginning of stoning with 3000 pilgrims. (**b**) During peak hour of stoning with 3000 pilgrims. (**c**) End of stoning with 3000 pilgrims. (**d**) Beginning of stoning with 7000 pilgrims. (**e**) During peak hour of stoning with 7000 pilgrims. (**f**) End of stoning with 7000 pilgrims. (**g**) Beginning of stoning with 10,000 pilgrims. (**h**) During peak hour of stoning with 10,000 pilgrims. (**i**) End of stoning with 10,000 pilgrims.

In the fire scenario, as shown in Figure 7, in all exit placements the percentage of evacuees drops significantly at the beginning of stoning (Figure 7a,d,g) compared to peak hours (Figure 7b,e,h) and the end of stoning (Figure 7c,f,i) phase. This observation confirms our previous reasoning, in Section 4.2, which states that in this scenario, most pilgrims are killed by the fire, and only a few survive and evacuate. This is because most pilgrims enter the area and are close to where the fire occurs.

In the case of a bomb explosion, Figure 8 shows that a high percentage of evacuees is achieved at the beginning of stoning. This could be because the bomb explosion starts at the beginning of the simulation and lasts only for 20 seconds and, hence, does not affect most pilgrims. Similar performance trends are observed during the peak hour and at the end of the phase.

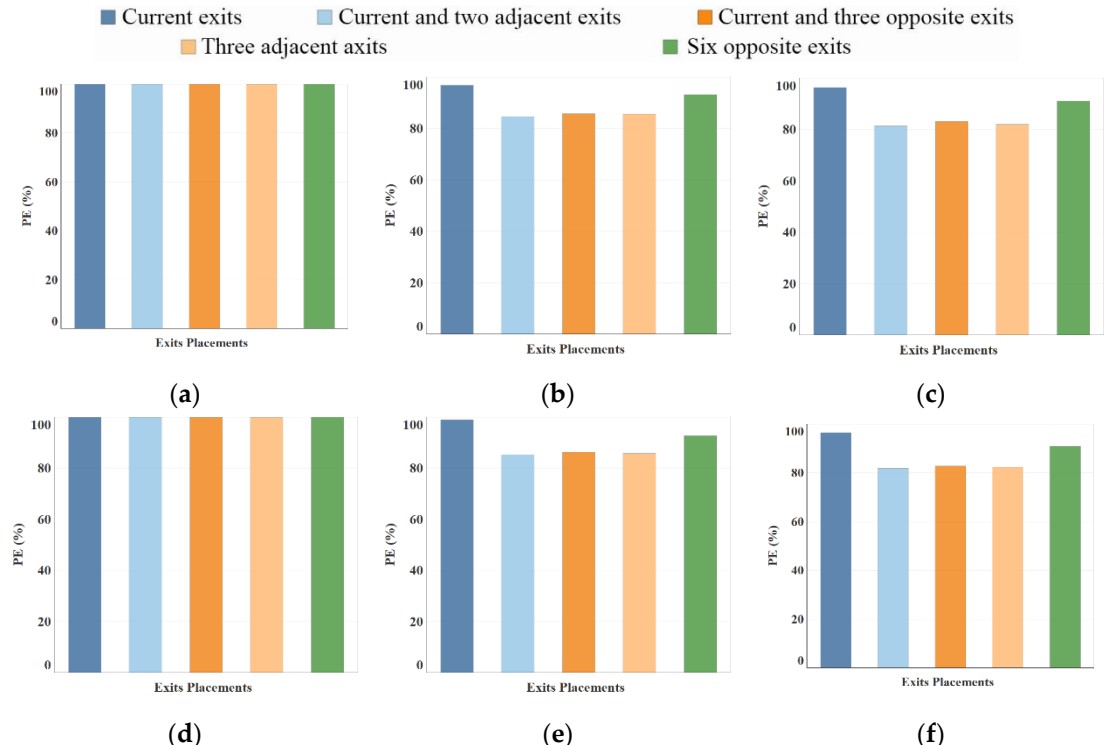

**Figure 8.** *Cont.*

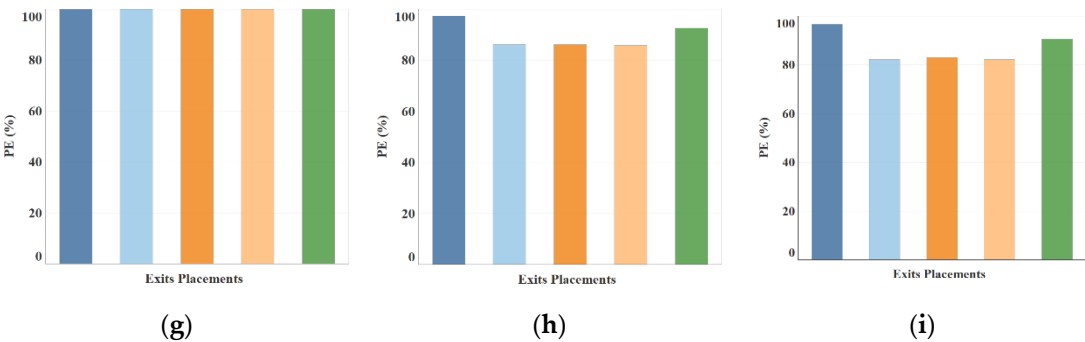

**(g)** **(h)** **(i)**

**Figure 8.** Percentage of evacuees (PE) for all scenarios in case of a bomb with 3000 pilgrims (**a**–**c**), 7000 pilgrims (**d**–**f**), and 10,000 pilgrims (**g**–**i**). (**a**) Beginning of stoning with 3000 pilgrims. (**b**) During peak hour of stoning with 3000 pilgrims. (**c**) End of stoning with 3000 pilgrims. (**d**) Beginning of stoning with 7000 pilgrims. (**e**) During peak hour of stoning with 7000 pilgrims. (**f**) End of stoning with 7000 pilgrims. (**g**) Beginning of stoning with 10,000 pilgrims. (**h**) During peak hour of stoning with 10,000 pilgrims. (**i**) End of stoning with 10,000 pilgrims.

### 4.4. Results on Percentage of Crowds at Exits

The Percentage of Crowds at Exits (PCE) measures the percentage of pilgrims who evacuate from each exit. The performance in respect to PCE is shown in Figures 9–11 with no hazard, fire hazard, and bomb hazard, respectively. In each figure, a sub-figure presents the PCE for each exit in each exit placement in a specific phase of stoning. For simplicity and space limitation, we only show results for a pilgrim population size of 7000 since other sizes share similar performance trends.

Figure 9 shows the case of no hazard where at the beginning of stoning (Figure 9a), all pilgrims use the closest exit to the entrance (exit 4) to evacuate since they are already close to the entrance, especially in the second, third, and fourth exit placements. In the sixth exit placement, i.e., six opposite exits, the majority of pilgrims (around 70%) use exit 4 to evacuate, while the rest use the second closest exit to the entrance, i.e., exit 6. In the first exit placement, i.e., current exits, more pilgrims use exit 2 than exit 1 since pilgrims need to travel a shorter distance to reach the exit.

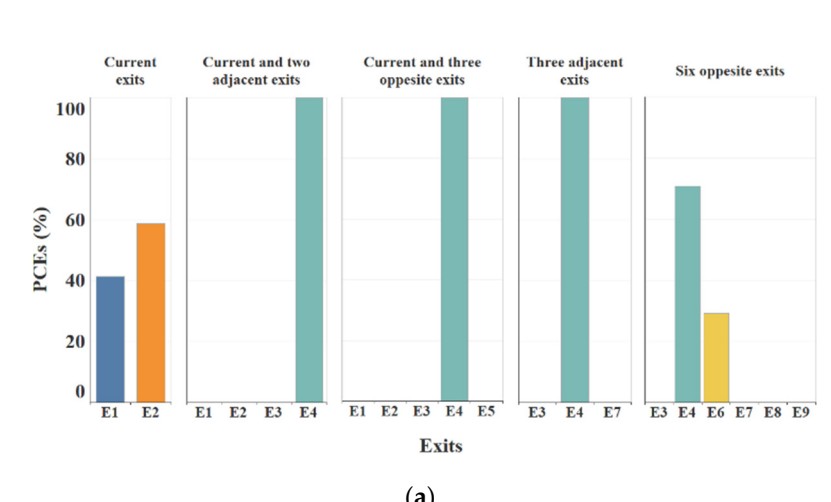

**(a)**

**Figure 9.** *Cont.*

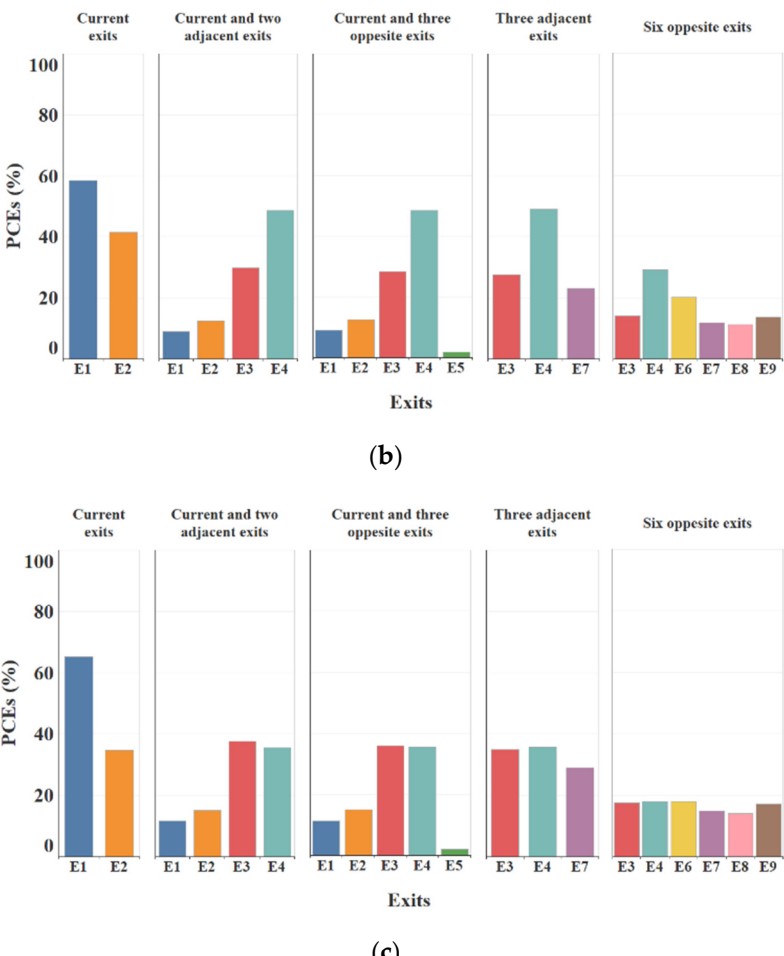

**Figure 9.** Percentage of the crowd at each exit for all scenarios in case of no hazard. (**a**) Beginning of stoning. (**b**) During peak hour of stoning. (**c**) End of stoning.

Similar trends are observed in the second phase of stoning, i.e., peak hour in Figure 9b, in the second, third, fourth, and fifth exit placements, where the majority of pilgrims use exit 4 to evacuate. Yet, more pilgrims are distributed across other exits since they are in the middle of the stoning area. The second most used exit in the second, third, and fourth exit placements is exit 3 because it is close to the second stone, while exit 6 is used more in the fifth exit placement as it is close to the entrance. In addition, more pilgrims are using exit 1 rather than exit 2. At the end of stoning, as shown in Figure 9c, the PCE across exits even out more, while still exits 3 and 4 are used by the majority of pilgrims. However, in all stoning phases, the current exits (exits 1 and 2) and the suggested exit 5 are minimally used for evacuation.

It is apparent that in the case of fire (Figure 10) most pilgrims are killed if it occurs at the beginning of stoning (Figure 10a), which is in agreement with our results in the previous Sections 4.1 and 4.2. Further, in case of fire (Figure 10b,c) and bomb (Figure 11), the superiority of exits 4 and 3 stands stronger as the majority of pilgrims use them in all stoning phases, while still exits 1, 2, and 5 are minimally used.

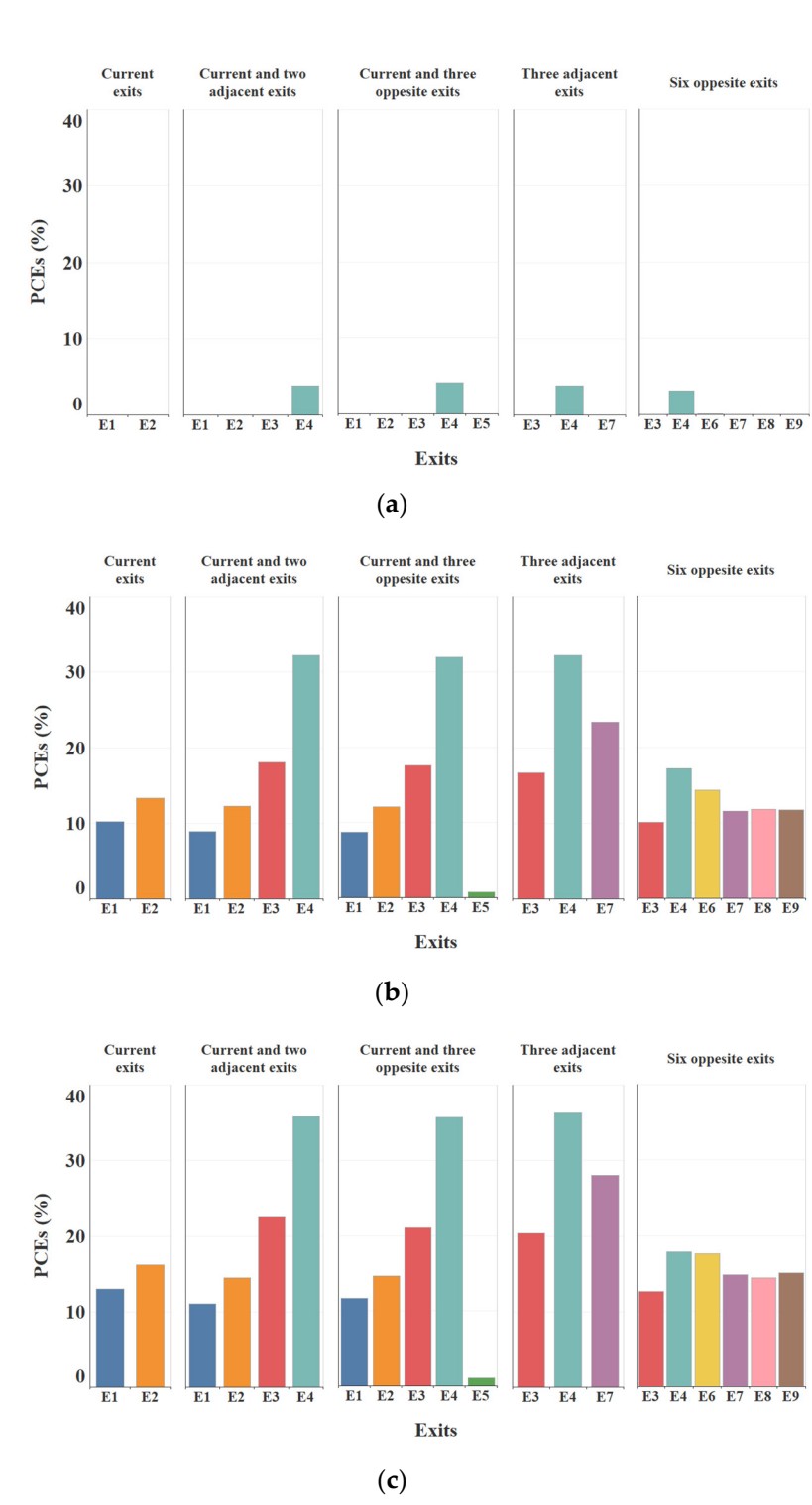

**Figure 10.** Percentage of the crowd at each exit for all scenarios in case of fire. (**a**) Beginning of stoning. (**b**) During peak hour of stoning. (**c**) End of stoning.

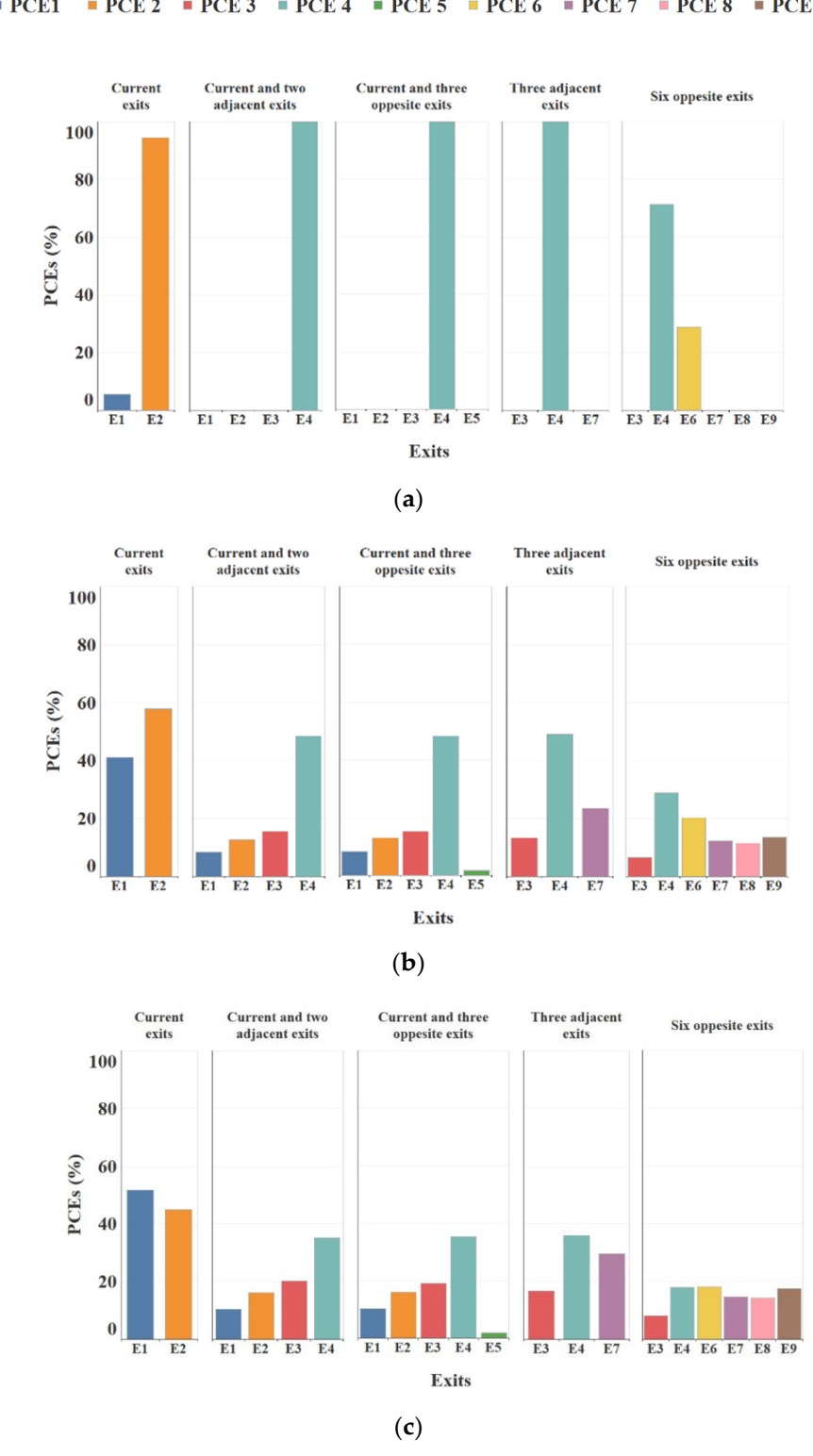

**Figure 11.** Percentage of the crowd at each exit for all scenarios in case of a bomb. (**a**) Beginning of stoning. (**b**) During peak hours of stoning. (**c**) End of stoning.

Taken together, these results suggest that the position of the current exits is not optimal since they were minimally used by the pilgrims in our simulation. Our results showed that the best positions for exits in the stoning area are near the entrance and next to each stone.

## 5. Conclusions and Future Work

This work aimed to investigate the evacuation effectiveness of the current exit placement in the stoning area of Hajj in Makkah, Saudi Arabia. To do so, we built a simulation model of the expanded stoning area with the current exit placement. We also modeled four different newly-proposed exit placements to compare with the current placement. Further, we considered evacuation in the case of three hazard situations namely no hazard, fire hazard, and bomb hazard. Furthermore, we analyzed evacuation during three different phases of stoning, at the beginning of stoning, during the peak hour of stoning, and at the end of stoning. We considered three different population sizes when measuring the performance taking into account the evacuation time, percentage of evacuees, and percentage of crowd at each exit.

The experimental results indicated that the current exit placements have a significantly longer evacuation time and a lower percentage of evacuees when compared to the proposed placements, specifically in the no-hazard and fire hazard scenarios. In such cases, our proposed exit placements could increase the percent of evacuees even more than three times compared to the conventional placements of the exits. Furthermore, through our rigorous empirical study, we show that the best positions for exits in the stoning area are near the entrance and next to each stone. These positions resulted in the best outcomes in most of the cases.

In the future, we plan to further extend this study by increasing the population size. We also intend to integrate more mobility models related to other disasters for exploring the behavior of the pilgrims in different other disastrous situations.

**Author Contributions:** Conceptualization, H.K. and A.B.M.A.A.I.; formal analysis, H.K., A.A. (Amal Alzuhair), D.A., H.A., N.A. (Noor Almoqayyad), R.A., A.A. (Alhanoof Althnian), N.A. (Najla Alnabhan) and A.B.M.A.A.I.; funding acquisition, N.A. (Najla Alnabhan); investigation, A.A. (Amal Alzuhair), D.A., H.A., N.A. (Noor Almoqayyad), R.A. and N.A. (Najla Alnabhan); methodology, H.K. and A.A. (Alhanoof Althnian); project administration, N.A. (Najla Alnabhan); software, A.A. (Amal Alzuhair), D.A., H.A., N.A. (Noor Almoqayyad) and R.A.; supervision, H.K.; validation, A.A. (Amal Alzuhair), D.A., H.A., N.A. (Noor Almoqayyad) and R.A.; visualization, A.A. (Amal Alzuhair), D.A., H.A., N.A. (Noor Almoqayyad), R.A. and A.A. (Alhanoof Althnian); writing—original draft, H.K., A.A. (Amal Alzuhair), D.A., H.A., N.A. (Noor Almoqayyad) and R.A.; writing—review and editing, H.K., A.A. (Alhanoof Althnian), N.A. (Najla Alnabhan) and A.B.M.A.A.I. All authors have read and agreed to the published version of the manuscript.

**Funding:** This work is supported by Deputyship for Research and Innovation, Ministry of Education in Saudi Arabia through the project number (DRI-KSU-762).

**Institutional Review Board Statement:** Not applicable.

**Informed Consent Statement:** Not applicable.

**Data Availability Statement:** Not applicable.

**Conflicts of Interest:** The authors declare no conflict of interest.

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
