# Peer review of "Crowd Evacuation in Hajj Stoning Area: Planning through Modeling and Simulation"

_sustainability, doi:10.3390/su14042278_

Round 1
Reviewer 1 Report
Thank you the submission of your manuscript to the Sustainabilityjournal. Generally, the manuscript fits into the scope of the journal,
snd the structure respects Scientific Best Practice. The topic is
interesting, and it was prepared in a comprehensive way. I do not have
too many comments. In the literature review, it is important that the scientific novelty of the work is established through a critical analysis of related literature.
Regarding the methodology: how did you do the validation of the model?
What are the uncertainties? In the conclusions, in addition to summarising the actions taken and results, please strengthen the explanation of their significance. It is recommended to use quantitative reasoning comparing with appropriate benchmarks, especially those stemming from previous work.
Reviewer 2 Report
This paper addresses the evacuation problem during Muslim pilgrinage to Hajj where crush accidents with many people killed are common.
Using Netlogo is a straightforward method for modelling and simulation of existing situation and propose alternatives.
The manuscript is well written and organized; no major or even minor aspects were found. Except, perhaps, a shortage of references, since there are so many, including seminal papers, that could be referenced. But it is well as it is, in my opinion. Introduction, Literature review (with the table) and method are well described, concise but complete.
Simulation results could be compared wih other similar experiments in order to evaluate and / or present different values. One question pops up my mind: is the population size used in the simulation consistent with the real scenarios? I would be nice to provid an annex (if possible) with the NetLogo code, typically very small in size.
Nevertheless, results are well presented and discussion is consistent with the simulation scenarios and data acquired. Well done.
